# Target Recognition Based on Infrared and Visible Image Fusion and Improved YOLOv8 Algorithm

**DOI:** 10.3390/s24186025

**Published:** 2024-09-18

**Authors:** Wei Guo, Yongtao Li, Hanyan Li, Ziyou Chen, Enyong Xu, Shanchao Wang, Chengdong Gu

**Affiliations:** 1School of Mechanical and Automotive Engineering, Guangxi University of Science and Technology, Liuzhou 545616, China; guo2008@foxmail.com (W.G.);; 2School of Automation, Guangxi University of Science and Technology, Liuzhou 545616, China; lihanyan@gxust.edu.cn; 3Dongfeng Liuzhou Motor Co., Ltd., Liuzhou 545616, China; chenzy@dflzm.com (Z.C.); xuey@dflzm.com (E.X.); wangsc@dflzm.com (S.W.)

**Keywords:** image fusion, attention mechanism, adaptive illumination perception, YOLOv8, median strengthening, spatial attention mechanism, channel attention mechanism

## Abstract

In response to the issue that the fusion process of infrared and visible images is easily affected by lighting factors, in this paper, we propose an adaptive illumination perception fusion mechanism, which was integrated into an infrared and visible image fusion network. Spatial attention mechanisms were applied to both infrared images and visible images for feature extraction. Deep convolutional neural networks were utilized for further feature information extraction. The adaptive illumination perception fusion mechanism is then integrated into the image reconstruction process to reduce the impact of lighting variations in the fused images. A Median Strengthening Channel and Spatial Attention Module (MSCS) was designed to be integrated into the backbone of YOLOv8. In this paper, we used the fusion network to create a dataset named *ivifdata* for training the target recognition network. The experimental results indicated that the improved YOLOv8 network saw further enhancements of 2.3%, 1.4%, and 8.2% in the Recall, mAP50, and mAP50-95 metrics, respectively. The experiments revealed that the improved YOLOv8 network has advantages in terms of recognition rate and completeness, while also reducing the rates of false negatives and false positives.

## 1. Introduction

The “New Four Modernizations” within China’s automotive sector—namely electrification, intelligence, interconnection, and sharing—are persistently advancing. Intelligence and interconnection are increasingly becoming standard features in new vehicles [1]. The auxiliary safety driving technology for vehicles in complex road environments has always been a hot topic in the automotive industry. It typically relies on Advanced Driver Assistance Systems (ADASs) to implement functions such as Adaptive Cruise Control (ACC), Lane Recognition, Forward Collision Warning (FCW), and Driver State Monitoring (DSM) [2].

An in-vehicle vision system has essentially become a standard feature in intelligent vehicles and is a crucial component of the ADAS. By employing visual cameras mounted on the vehicle’s body to detect and recognize the external environment, and by performing real-time analysis of the collected environmental information, it is possible to achieve recognition of surrounding vehicles, driving lanes, traffic signs, and pedestrians. This, in turn, enables functions such as Lane Departure Warning (LDW), FCW, Pedestrian Detection [3], Traffic Sign Recognition (TSR), and Intelligent High Beam Control (IHBC) [4].

Visible images and infrared images have complementary characteristics. Visible images can provide rich color and textural information, but they are very sensitive to lighting conditions and do not perform well in night-time or low-light environments [5]. Visible images capture the spectral details reflected by objects, providing rich textural information, as shown in Figure 1a, which is not available in infrared images [6]. In contrast, infrared images can capture thermal radiation information, are not affected by lighting changes, and can provide image information under night-time or adverse weather conditions, as shown in Figure 1b. Infrared imaging primarily relies on the thermal characteristics of objects rather than their surface features, which means infrared images have limitations and lack the textural and detailed richness found in visible images [7]. Fusing these two types of images can generate images that contain more information, enhancing the usability and application range of the images. The fusion of infrared and visible images can highlight environmental targets while preserving some details in low-light areas, such as the trash can shown in Figure 1c. This image fusion technology has been widely applied to various fields, including military [8], target recognition [9], tracking [10], and semantic segmentation [11].

With the advancement in deep learning, deep learning-based fusion techniques have gradually taken a dominant position in the field of infrared and visible image fusion. Li et al. proposed DenseFuse [12], featuring an encoding layer with convolutional and dense blocks that maximize the retention of deep features and integrate all prominent features into the fusion process. Ma et al. proposed the FusionGAN [13] method using Generative Adversarial Networks(GANs), where one network adds texture to infrared images and another discerns the authenticity of the fused results. Zhang et al. proposed IFCNN [14], which utilizes convolutional neural networks to integrate features from multiple input images into a comprehensive single image, allowing for the optimization of all model parameters without the need for post-processing. Zhang et al. proposed PMGI [15], which introduces a Pathwise Transfer Block to exchange information between different paths. This not only allows for the preliminary fusion of gradient and intensity information but also enhances the information to be processed subsequently. Xu et al. proposed U2Fusion [16], a method that assesses and fuses image information based on feature extraction and importance. It is a non-end-to-end approach, using a series of models where each model’s output feeds into the next.

During the fusion of infrared and visible images, illumination often affects the quality of fusion. To address the lighting issue, Tang et al. proposed a fusion method called DIVFusion [17], which incorporates the low-light enhancement algorithm Retinex into the fusion network to increase the brightness of visible light images, thereby improving the brightness of the fused images. Tong et al. proposed a three-stage fusion algorithm for infrared and visible image fusion under different illumination conditions: image preprocessing, multi-scale decomposition, and image fusion [18]. Yang et al. proposed a framework that includes an illumination enhancement network and an image fusion network. The network incorporates two novel modules, the Salient Target Aware Module (STAM) and the Adaptive Differential Fusion Module (ADFM), which enhance the quality of a fused image [19]. When there is strong light exposure in the environment, as shown in Figure 1d, the visible image is in a high brightness state. Continuing to increase this brightness would result in too little infrared information being retained in the fused image, as shown in Figure 1f, which is not conducive to subsequent target recognition. To adapt the fusion network to the impact of different lighting factors on the brightness of visible light images in nocturnal environments, in this paper, we propose an adaptive illumination perception fusion mechanism, which was incorporated into the image reconstruction phase of the fusion network. In nocturnal environments, when there is strong light interference in a visible light image, it can still adaptively adjust the fusion ratio, retain more infrared information, and facilitate subsequent target recognition.

After the fusion of infrared and visible images, the fused image contains new feature information. Using YOLOv8 for direct object detection results in high false-positive and false-negative rates. It is necessary to improve YOLOv8 to enhance the network’s feature extraction capabilities. In this field, Duan et al. proposed a dual-channel salient object recognition algorithm based on an improved YOLO network [20]. Wu et al. first performed preprocessing on dual-modality images, followed by the quantitative analysis of the images [21]. For this paper, we designed an attention module, integrated into the backbone of the YOLOv8 network. Through experiments, we will prove that the improved YOLOv8 network in this paper has higher accuracy in target recognition for fused images.

To develop an in-vehicle target recognition system based on infrared and visible image fusion, the main contributions of the paper are as follows:In the fusion network, a spatial attention mechanism enhances feature extraction, preserving detailed textures. Depthwise separable convolutions streamline the process to boost fusion speed.In the infrared and visible image fusion network, an adaptive illumination perception fusion mechanism was added to adaptively adjust the brightness of the fused image.A Median Strengthening Channel and Spatial Attention Module (MSCS) were designed, integrating the MSCS module into the backbone of the YOLOv8 network, enhancing the network’s recognition accuracy and real-time performance.The ivifdata dataset was created using the fusion network presented in this paper to train both the YOLOv8 network and the improved YOLOv8 network.

The remainder of this paper is organized as follows. Section 2 briefly describes the related works on infrared and visible image fusion and the illumination perception fusion mechanism. In Section 3, comparative experiments were conducted with other fusion methods, and the results are visually presented in the forms of tables and line charts. In Section 4, the detailed process of improving YOLOv8 is illustrated. In Section 5, improved YOLOv8 performance is illustrated through experiments, followed by some concluding remarks in Section 6.

## 2. Infrared and Visible Image Fusion

### 2.1. Fusion Network

The algorithm’s structure is depicted in Figure 2, and the specific steps are the following:During the feature extraction phase, an infrared image is first directly processed through the spatial attention mechanism. To reduce computational complexity, the visible image is preprocessed through a depthwise separable convolution layer before also being input into the spatial attention mechanism. In this way, spatial attention weight maps can be obtained. These weight maps are then multiplied with the original images to enhance spatial salient features. These enhanced features are then sent into a convolutional neural network (CNN) to further extract deep-level features.The deep features obtained from the CNN are integrated to form the input data for the image reconstruction process. An improved illumination aware weight fusion module is incorporated during the image fusion process to enhance the adaptability of the fusion network, preserving more information features for the fused image, ultimately generating the final fused image.Utilizing the weights obtained from the infrared and visible images, a loss function is constructed. This loss function will guide the training process, optimizing the network parameters through backpropagation to enhance the model’s performance.

### 2.2. Attention Mechanism

In visual tasks, attention mechanisms are commonly used to enhance the performance of models and their sensitivity to feature information in images. Common types include spatial attention, channel attention, hybrid attention, and self-attention mechanisms [22]. Spatial attention mechanisms focus on the spatial dimensions of an image, that is, the width and height of the image. They mimic the human visual system’s ability to focus on specific areas of an image, processing the region most relevant to the task, concentrating on the areas where the salient targets are located in the image to be processed, and emphasizing the handling of detailed textural information [23].

The spatial attention module used in this paper is shown in Figure 3, and its operation process is as follows.

The input image first undergoes average pooling and max pooling to obtain two feature descriptors. These descriptors are concatenated along the channel dimension to form a comprehensive feature tensor. Subsequently, this tensor passes through a convolutional layer with a 7 × 7 kernel, and finally, a nonlinear mapping through the Sigmoid activation function is applied to generate a spatial attention map. In the illustrations within the text, “Average Pooling” and “Max Pooling” represent the operations of average and max pooling, respectively; “Concatenation” indicates the merging of the two feature tensors along the channel dimension. The design of this spatial attention module aims to capture key information in an image through precise feature fusion and nonlinear transformation, thereby providing an enhanced feature representation for subsequent visual tasks.

### 2.3. Utilizing Depthwise Separable Convolution to Simplify Computations

Factorized Convolution [24] is an efficient convolutional operation technique, as shown in Figure 4, consisting of two consecutive convolution processes: the first is depthwise convolution, followed by pointwise convolution. Compared to traditional convolution, depthwise separable convolution significantly reduces the number of parameters while maintaining the same output feature map [25].

Depthwise separable convolution splits standard convolution into two phases: depthwise and pointwise convolution. During the depthwise convolution, each input channel is independently processed by a specific kernel, focusing on extracting local spatial features for that channel without increasing the number of feature maps. This approach allows the model to concentrate on learning channel-specific features while avoiding the mixing of information between different channels. However, it also restricts the integration of information from different channels at the same pixel location. To address this, the pointwise convolution phase follows, where 1 × 1 kernels are used to mix the feature maps generated by the depthwise convolution, enabling cross-channel feature integration. This two-step process reduces the computational complexity and parameter count while refining and enhancing the expressiveness of the feature representations.

### 2.4. Illumination Perception Weight Fusion Module

The Illumination Perception Weight Fusion (IPWF) module is a technique used for multimodal image fusion, particularly in the domain of visible and infrared image fusion. This module is designed to adaptively adjust the fusion weights of different modal images by considering the image characteristics under varying lighting conditions, thereby enhancing the quality of the fused image. The module’s structure is shown in Figure 5. Considering the limitations of infrared images in capturing environmental lighting changes, visible features are used as the input for the IPWF network to achieve image fusion under varying lighting conditions.

The weight distribution for infrared and visible images during the fusion process is based on the probabilistic outputs provided by the IPWF module.
(1)$$W_{V}=\frac{p_{d}}{p_{d}+p_{n}}$$
(2)$$W_{I}=\frac{p_{n}}{p_{d}+p_{n}}$$
where $p_{d}$ is the probability of using visible images captured during the daytime for fusion, and $p_{n}$ is the probability of visible images being captured at night. When the visible image is input into the IPWF module, it yields the values $p_{d}$, $p_{n}$. $W_{V}$ represents the weights of the visible image in the loss function, and $W_{I}$ represents the weights of the infrared image in the loss function.

After enhancement by the attention mechanism module, the visible and infrared features are input into the proposed IPWF module for weight adjustment and fusion.

The IPWF module consists of a 3 × 3 convolutional layer, an SKNet layer [26], and a Softmax function, which together form the weight generation part. As shown in Figure 6, the SKNet network can dynamically generate various convolutional kernels based on the specific features of the image and then select the most suitable convolutional kernel for feature extraction for the current image.

The SKNet layer consists of three steps: segmentation, fusion, and selection.

Segmentation. The input feature map undergoes 3 × 3 and 5 × 5 convolutional operations, resulting in two feature maps.Fusion. By adding $U_{1}$ and $U_{2}$ together, U is obtained, which is a feature map that encompasses features from different receptive fields.



(3)
$$U=U_{1}+U_{2}$$



$S_{C}$ is a feature factor that contains global information, obtained through a Global Average Pooling (GAP) operation.
(4)$$S_{C}=F_{GAP}\left(U\right)=\frac{1}{H\times W}\sum_{i=1}^{H}\sum_{j=1}^{W}U\left(i,j\right)$$
where H and W represent spatial dimensions in Formula (4).

Through a linear mapping by a fully connected (FC) layer, a dimensionality-reduced feature representation z is obtained, which helps to improve the operational efficiency of the model.
(5)$$z=F_{FC}\left(s\right)=\delta \left[\beta \left(W_{s}\right)\right]$$
where δ represents the ReLU activation function, z represents the feature, and β represents the Batch Normalization (BN) layer in Formula (5).

3.Selection. Through the fusion process, the system can automatically adjust and capture weight information at different spatial scales. Subsequently, the Softmax function is applied to the variable z to obtain the weight distribution for feature layers a and b. The calculation formula is shown as Formula (6).(6)$$\left\{\begin{array}{c} a=\frac{e^{Az}}{e^{Az}+e^{Bz}}\\ b=\frac{e^{Bz}}{e^{Az}+e^{Bz}} \end{array}\right.$$

Continuing with the weighted calculation, the vector V is obtained, as shown in Formula (7).
(7)$$\left\{\begin{array}{c} a+b=1\\ V=aU_{1}+bU_{2} \end{array}\right.$$

The SKNet layer optimizes the features of visible images, emphasizing the feature representation of the targets while suppressing the interference from the background and other non-critical features. The refined feature information is then fed into a fully connected layer, which analyzes these features to predict the weights of the images under various lighting conditions, distinguishing between day and night scenes.

The Softmax function can transform a vector or a set of real numbers into a probability distribution, ensuring that each element’s value lies between 0 and 1, and the sum of all elements equals 1. By placing the Softmax function after the SENet layer, the model’s raw output is converted into a weighted output, as expressed in Formula (8).
(8)$$Softmax\left(z_{i}\right)=\frac{e^{z_{i}}}{\sum_{j}e^{z_{j}}}$$
where $z_{i}$ represents the raw score of the i-th output from the model, the numerator $e^{z_{i}}$ is the exponentiation of $z_{i}$, and the denominator is the sum of all $e^{z_{j}}$, which ensures that the sum of all output probabilities equals 1.

#### Adaptive Illumination Perception Fusion Mechanism

Existing illumination weighting fusion strategies, while capable of automatically adjusting the weights of visible and infrared features according to the lighting conditions of the day and night, achieving adaptive feature fusion, have limited adaptability.

As shown in Figure 7a,b, the shooting environment is at night, and the brightness of the image is increased due to the illumination of the car lights. The fusion network will increase the fusion weight of visible light during the fusion process, but increasing the proportion of infrared images in the fusion process would yield better results. As shown in Figure 7c,d, if there is a highlight in the visible light image in the night environment, the weight of visible image will be increased. As a result, the two pedestrians obscured by the car will have reduced weight in the fusion process, leading to a deviation from the expected fusion effect. To reduce the interference of the above issues on fusion, in this paper, we propose an improved adaptive illumination fusion mechanism. By readjusting the day and night weights extracted from the visible image, more universal weights for visible and infrared features are obtained. On this basis, further weighted fusion of features is carried out.

In the IPWF, adjustments were made to $p_{d}$, $p_{n}$ to obtain $W_{I}^{\prime }$, $W_{V}^{\prime }$, respectively. The adjustment formula is shown as Formula (9):(9)$$\left\{\begin{array}{c} W_{I}^{\prime }=b\cdot \frac{p_{d}-p_{n}}{2}+\frac{1}{2}\\ W_{v}^{\prime }=1-W_{I}^{\prime }\\ b=\alpha_{\omega }\cdot \left|\omega \right|+\gamma_{\omega } \end{array}\right.$$

To ensure that the weight values of visible and infrared features more accurately reflected the characteristics of their respective images, the initial weights for both were first set to the same value, that is, 0.5. Then, starting from the base weight of 0.5, the differences between the weights were learned and adjusted. Here, b represents the deviation in the weight, $0\leq \left|\omega \right|\leq 1$, and $\alpha_{\omega }$, $\gamma_{\omega }$ were set to 0 and 1, respectively, as learnable parameters.

The visible weight values obtained by applying the Sigmoid function were refined to enhance the robustness of the illumination mechanism, thereby deriving $W_{I}$. The Sigmoid operation is denoted by θ, and the formulas are expressed as Formulas (10) and (11):(10)$$\theta \left(x\right)=\frac{1}{1+e^{-x}}$$
(11)$$\left\{\begin{array}{c} W_{I}=\theta \left(W_{I}^{\prime }\right)\\ W_{V}=1-W_{I} \end{array}\right.$$

Finally, the weight values obtained for the visible and infrared images were individually applied to their corresponding original features. By employing multiplication and cascading operations, the ultimate fused image features were synthesized.

### 2.5. Loss Function

#### 2.5.1. Fusion Network Loss Function

Introducing an illumination perception loss function enables a fusion network to adaptively adjust under various lighting conditions, integrating more meaningful feature information. The formula for the light-aware loss function is Formula (12):(12)$$L_{i}=W_{I}\cdot L_{loss}^{I}+W_{V}\cdot L_{loss}^{V}$$
where $W_{I}$ and $W_{V}$ represent the perceptual weights for the infrared and visible light images, respectively. $L_{LOSS}^{I}$ and $L_{LOSS}^{V}$ denote the $L_{LOSS}$ for the images, and the light-aware loss function can be formulated to adapt a fusion network to different lighting conditions and integrate more meaningful feature information. The differences between the input and output images can be computed at the pixel level, with Formulas (13) and (14):(13)$$L_{LOSS}^{I}=\frac{{\left\|I_{F}-I_{I}\right\|}_{1}}{HW}$$
(14)$$L_{LOSS}^{V}=\frac{{\left\|I_{F}-I_{V}\right\|}_{1}}{HW}$$
where H refers to the number of pixels in the height of the image, and W refers to the number of pixels in the width of the image. ${\left\|\cdot \right\|}_{1}$ denotes the norm of the $l_{LOSS}$.

To adapt to different lighting environments, a light-aware weight fusion module loss function was designed to dynamically maintain the brightness information of the source images. However, relying solely on the light-aware loss is not sufficient to ensure that a fused image achieves the desired brightness balance. Therefore, it was necessary to introduce an auxiliary intensity loss function, which could further optimize the brightness distribution of the fused image to achieve a more natural and consistent visual appearance, with the formula being Formula (15):(15)$$L_{a}=\frac{{\left\|I_{F}-max\left(I_{I},I_{V}\right)\right\|}_{1}}{HW}$$

Furthermore, the fused image should not only maintain an ideal intensity distribution but also preserve rich textural details, which are crucial for the visual effect of the image. To this end, this study introduces a texture loss constraint aimed at enhancing the textural features of the fused image. Considering that the best textural features are often manifested as a combination of the texture features of infrared and visible light images, in this study, we adopted a fusion strategy based on textural maximization to ensure that the fused image could integrate the textural information from both image sources, thereby achieving a richer and more delicate visual effect. The formula for the textural detail loss $L_{t}$ is Formula (16):(16)$$L_{t}=\frac{{\left\|\left|{\nabla I}_{F}\right|-max\left(\left|{\nabla I}_{I},{\nabla I}_{F}\right|\right)\right\|}_{1}}{HW}$$
where ∇ represents the Sobel gradient operator used to measure the textural information of the image, and $\left|\cdot \right|$ denotes the absolute value operation.

The global loss function of the fusion network is shown as Formula (17):(17)$$L_{f}=\mu_{1}L_{i}+\mu_{2}L_{a}+\mu_{3}L_{t}$$

#### 2.5.2. IPWF Module Loss Function

The quality of a fused image largely depends on the accuracy of the illumination-aware weight fusion network. Essentially, the network plays the role of a classifier, whose task is to assess the probability of an image belonging to a daytime or night-time scene. To ensure the accuracy of the network training, a cross-entropy loss function *L_N_* was used to guide the training process. The cross-entropy loss function can effectively quantify classification errors and optimize network parameters through a backpropagation algorithm, thereby improving the network’s accuracy in classifying images under different lighting conditions. The formula is shown as Formula (18):(18)$$L_{N}=-\rho \log\varphi \left(y\right)-\left(1-\rho \right)\log\left(1-\varphi \left(y\right)\right)$$
where ρ represents the illumination label of the source image, and $y=\left(p_{d},p_{n}\right)$, where the Softmax function is denoted by φ(·), which normalizes y to normalize the illumination probability to [0,1].

## 3. Infrared and Visible Image Fusion Experiment

### 3.1. Experimental Description

To validate the performance of the fusion network presented in this paper during the process of fusing infrared and visible images, multiple pairs of infrared and visible images were selected from public datasets including TNO (available at https://figshare.com/articles/dataset/TNO_Image_Fusion_Dataset/1008029, accessed on 3 March 2024), LLVIP (available at https://bupt-ai-cz.github.io/LLVIP, accessed on 10 March 2024), RoadScene (available at https://github.com/hanna-xu/RoadScene, accessed on 11 March 2024), and Multi-Spectral Road Scenarios (MSRS, available at https://github.com/Linfeng-Tang/MSRS, accessed on 15 March 2024) for experimentation. Additionally, this paper’s algorithm was compared with five representative fusion algorithms, namely DenseFuse, FusionGAN, IFCNN, PMGI, and U2Fusion.

Among them, DenseFuse is a generative method that employs pixel-level dense fusion technology; FusionGAN is a method that achieves image fusion using a Generative Adversarial Network (GAN); IFCNN achieves efficient image fusion through convolutional layer fusion and convolutional layer reconstruction; the PMGI method focuses on achieving fusion effects by maintaining the ratio of gradients and intensities; and U2Fusion automatically assesses the importance of source images and performs fusion through feature extraction and information measurement techniques. During the comparative experimental process, all the comparative experimental setups were trained according to the open-sourced code without modifying any parameters. The experimental equipment configuration and environmental setup are shown in Table 1.

### 3.2. Experimental Details

Deep learning networks necessitate extensive training datasets, and for this study, we chose to construct the training dataset based on the MSRS dataset. From this dataset, we selected 427 daytime scene images and 376 night-time scene images. These images required cropping prior to training, with a stride of 64, yielding images cropped to a size of 64 × 64 pixels. This resulted in a total of 29,960 daytime image patches and 26,320 night-time image patches. Furthermore, we utilized 376 pairs of daytime images and 376 pairs of night-time images, amounting to 26,320 image patches for each, to train the parameters of the fusion network. Before the image patches were fed into the network, they underwent standardization. Throughout the training process, we employed a single thermal target label to guide the training of the illumination-aware network. Specifically, the daytime scene label was set to [1,0], while the night-time scene label was set to [0,1].

#### 3.2.1. Comparative Analysis

Firstly, experiments were conducted on the selected algorithms and the algorithm presented in this paper under daytime conditions with ample sunlight. The infrared and visible test images were selected from the MSRS dataset, specifically image 00556D. The experimental results are shown in Figure 8.

According to the results in Figure 8, the algorithm presented in this paper retains more details while highlighting the human body target. Based on the detailed information shown within the red and green boxes, due to the introduction of spatial attention mechanisms in the fusion network, the overall brightness of the fusion result from this paper’s algorithm is moderate, and the visual perception is more natural. The regions selected in the fusion images of DenseFuse and IFCNN are too dark. FusionGAN and PMGI, on the other hand, result in overexposure, leading to blurred edge information in the framed areas. Although U2Fusion achieves a moderate brightness in the fused image, the infrared information of the human target in the fusion result is lost, making the human body less clear compared to the fusion result obtained by the algorithm in this paper.

In addition to the comparisons conducted in the daytime scenarios of the MSRS dataset, in this paper, we also carried out comparative experiments in the night-time scenarios of the MSRS dataset, as shown in Figure 9.

Based on the test results from the night-time scenarios, the method presented in this paper fully retains human bodily information without causing information pollution. At the same time, the brightness of the fused image is appropriate, without overexposure, and the edges of the detailed information are clear.

The generalization performance of deep learning algorithms is an important metric for measuring the performance of an algorithm. After training on the dataset, the algorithm presented in this paper was directly tested on the TNO dataset and the RoadScene dataset. Four sets of images were selected for testing, and the fusion results are shown in Figure 10, Figure 11, Figure 12 and Figure 13.

Based on the results in Figure 10, the images from DenseFuse and IFCNN exhibit lower brightness, while those from FusionGAN and PMGI are overly bright, with the edges of the figures in the red frame appearing rough. U2Fusion has a suitable overall brightness, but the figures are not as clear after fusion as with the method presented in this paper. In Figure 11, this paper’s method ensures that the details in the fused image are preserved, with the figures retaining enough infrared information to appear clearer. In the results of Figure 12, the visual appeal is best for DenseFuse, U2Fusion, and the method of this paper. The fusion results of FusionGAN and PMGI are severely overexposed due to the influence of visible brightness, and IFCNN has a noticeable color bias in the fusion. In the fusion results of Figure 13, FusionGAN and PMGI are again too bright, leading to a loss of detail. The algorithm in this paper, which incorporates an attention mechanism, retains more details of the subjects during the fusion process, making the human targets more prominent in the fused images.

For the adaptive illumination-aware fusion mechanism mentioned in this study, data were selected from the LLVIP dataset for experimentation. The experimental results are shown in Figure 14, Figure 15, Figure 16 and Figure 17.

From the experimental results mentioned above, it is evident that the method in this paper preserves ample infrared information in the fusion results for night scenes with low light, resulting in clear figure information. In Figure 15, the figures within the green frame are in an area of high brightness. FusionGAN and IFCNN are affected by the brightness of the visible image, retaining too much visible information and losing infrared information. The fusion results of DenseFuse and PMGI have uneven brightness, and U2Fusion introduces redundant information, leading to some details being lost and blurred. Using the fusion algorithm with the adaptive illumination perception fusion mechanism added in this paper for image fusion, the resulting fused images can still retain more infrared features even under strong night-time lighting conditions. In the comparison of the fusion results in Figure 14 and Figure 16, the method of this paper ensures the retention of sufficient infrared information in scenes with low visible image brightness. In the comparison in Figure 17, the visible image scene is at night, but there is high light interference in the visible image. The method of this paper adaptively adjusts the weights of infrared and visible information, ensuring that the scenes at night with high light interference still primarily retain more infrared information.

#### 3.2.2. Quantitative Analysis

Six evaluation metrics were selected for the quantitative analysis of the fused images: Mutual Information (MI), Visual Information Fidelity (VIF), Average Gradient (AG), Fused Image Quality (Qabf), Entropy (EN), and Spatial Frequency (SF). Mutual Information measures the degree of detail retention between the source and fused images; the larger the MI, the more source image details are retained in the fused image. The five metrics of VIF, AG, Qabf, EN, and SF were all used to assess the amount of information, significance, grayscale variation, and clarity contained in the fused image. The higher the values of these six metrics, the better the quality of the fused image.

An objective evaluation analysis was conducted based on 42 pairs of infrared and visible images from the TNO dataset. The data presented in Table 2 are the average values for each fusion method, and a line chart of the test data is shown in Figure 18. The algorithm of this paper showed the best performance in four evaluation metrics: MI, VIF, Qabf, and SF. This signifies that the images fused by the algorithm in this paper retained the most information from the original images, offered the highest visual quality, and also had the most pronounced boundary features in the fused images.

The comparative experimental results of this study demonstrated that the proposed method not only effectively retained the characteristics of infrared targets during the fusion process but also significantly avoided the textural detail retention issues caused by uneven brightness in the images. Compared to existing technologies, this method shows a marked improvement in maintaining the integrity of image information and enhancing visual effects, with an overall superior performance.

## 4. Deep Learning-Based Object Recognition

YOLOv8 is a single-stage detection algorithm that integrates state-of-the-art (SOTA) techniques based on the YOLOv5 algorithm, enhancing its capabilities. The YOLOv8 model comes in different versions based on network depth and width, namely YOLOv8n, YOLOv8s, YOLOv8m, YOLOv8l, and YOLOv8x. For this study, we selected the YOLOv8n network, which has a smaller size, higher precision, and faster running speed for optimization and improvement [27]. The network architecture of YOLOv8 is shown in Figure 19.

The YOLOv8 model’s structure mainly consists of four parts: the Input, Backbone, Neck, and Head. The Input end is responsible for the necessary preprocessing of the input image to meet the requirements of model training. This includes adjusting the image size to conform to the predetermined specifications, performing image scaling to optimize resolution, adjusting the color balance of the image to enhance visual effects, and applying the Mosaic data augmentation technique to improve the model’s generalization ability.

The experiments of our study were based on the YOLOv8n model, characterized by a smaller parameter count and higher precision. Further exploration of the model’s enhancement methods is pursued within the context of infrared and visible image fusion, with the expectation of reaching a higher level of performance in applications involving the fusion of infrared and visible images, which differ from conventional scenarios.

### Improving the YOLOv8 Network

The targets from the fusion of infrared and visible images have distinct features compared to those in the visible images. At night, the fused targets preserve many characteristics from the infrared images. During the day, the fused images retain features from both the infrared and visible images, which can be seen as having multi-scale features. In the research of deep learning-based machine vision, the performance of network models heavily relies on the ability of an algorithm to extract features of the targets in the images. Although traditional CNNs are proficient at capturing local features of images, they face certain constraints in dealing with global information and multi-scale features. Additionally, the key focus in this study’s improvement in the YOLOv8 network was how to enhance the model’s noise robustness while keeping computational efficiency.

We propose an innovative spatial and channel attention module named a Median Strengthening Channel and Spatial Attention Module (MSCS), which combines channel attention and spatial attention mechanisms. The MSCS module is shown in Figure 20.

The module was designed to improve the efficiency and robustness of feature extraction. In terms of channel attention, global pooling operations are used to detect and integrate global statistical features. Spatial attention is achieved through multi-scale depth convolution, capable of capturing spatial features at various scales, thus enriching the diversity of the features. The MSCS module’s overall architecture aims to offer a more holistic view, enhancing the model’s ability to express complex data and, consequently, improving overall performance.

The channel attention mechanism aggregates the global statistical information of the input feature maps to generate a channel attention map, thereby weighting the channels of the input features. The specific process is as follows.

Pooling Operation

Global Average Pooling (AvgPool), Global Max Pooling (MaxPool), and Global Median Pooling (MedianPool) are performed on the input feature maps to obtain three distinct pooling results. Each of the pooling results has the following size:$$R^{C\times 1\times 1}$$
where C represents the number of channels.

2.Shared MLP Processing

Each pooling result is processed through a shared Multilayer Perceptron (MLP), which includes two 1 × 1 convolutional layers and a ReLU activation function. The first convolutional layer reduces the feature dimension from (C) to (C/r), where (r) is the reduction ratio, and the second convolutional layer restores the feature dimension to (C). Finally, a Sigmoid activation function is applied to compress the output values into the range [0, 1], resulting in three attention maps.

3.Fusing Pooling Results

The attention maps from the three pooling results are element-wise added together to obtain the final channel attention map.

4.Generating the Spatial Attention Map

The channel attention map is multiplied with the original input feature map on an element-wise basis to obtain the weighted feature map. The corresponding formulas are shown as Formulas (19) and (20):(19)$$F_{C}=\sigma \left(MLP\left(AvgPool\left(F\right)\right)\right)+\sigma \left(MLP\left(MaxPool\left(F\right)\right)\right)+\sigma \left(MLP\left(MedianPool\left(F\right)\right)\right)$$
(20)$$F^{\prime }=F_{C}⊙F$$

In Formulas (19) and (20), σ represents the Sigmoid function, and ⊙ denotes element-wise multiplication.

The spatial attention mechanism captures the spatial relationships of the input feature maps through multi-scale depth convolution and generates a spatial attention map. The specific process is as follows.

Initial Convolutional Layer

The input feature map first passes through a 5 × 5 depth convolutional layer to extract basic features. The output size of this convolutional layer is the same as the input.

2.Multi-Scale Depth Convolution

The output feature map from the initial convolutional layer is then passed through multiple depth convolutional layers of various sizes, including convolutions of different dimensions, such as 1 × 11, 1 × 7, etc., to further extract multi-scale features.

3.Feature Fusion

The outputs from all the depth convolutional layers are added together element-wise to obtain the fused feature map.

4.Generating the Spatial Attention Map

The fused feature map is processed through a 1 × 1 convolutional layer to generate the final spatial attention map. This attention map is then multiplied element-wise with the channel-weighted feature map to obtain the final output feature map. The corresponding formulas are shown as Formulas (21) and (22):(21)$$F_{s}=\sum_{i=1}^{n}D_{i}\left(F^{\prime }\right)$$
(22)$$F^{\prime \prime }=Conv1\times 1\left(F_{s}\right)⊙F^{\prime }$$

In Formulas (21) and (22), $F_{s}$ represents the depth convolution operation of different sizes, n denotes the number of depth convolutions, and Conv1×1 indicates the 1 × 1 convolution operation.

Integrating the MSCS module into the tail of the YOLOv8 backbone network coordinates the distribution of attention in a gentler manner, reducing the network’s sensitivity to sudden changes and increasing its sensitivity to image feature information. The improved YOLOv8 network architecture is shown in Figure 21.

## 5. Improved YOLOv8 Network Experiments and Result Analysis

### 5.1. Dataset Preparation

The YOLOv8 network model was used to identify targets in the fusion of infrared and visible images, mainly for the recognition of vehicles, pedestrians, and other objects in vehicle-mounted infrared and visible fusion information during field experiments. To better develop the performance of the network model, an infrared and visible fusion image dataset called ivifdata was created using publicly available infrared and visible datasets.

The infrared and visible fusion image dataset mentioned in this paper, named ivifdata, sources its infrared and visible images from the publicly available LLVIP dataset. A total of 2000 pairs of images were selected and fused using the fusion algorithm proposed in this study. The selected pairs of images included a variety of categories such as pedestrians, bicycles, motorcycles, cars, buses, trucks, animals, and more, totaling over a dozen types. Manual annotation was performed on these images using the image annotation tool LabelImg. Additionally, 500 pairs of fused images were chosen to serve as the test set. The dataset annotation is shown as Figure 22.

### 5.2. Experimental Environment

The YOLOv8 experiments in this study were conducted on a computer equipped with the Windows 11 operating system. The version of Python used was 3.11, and the PyTorch version was 2.3.1. Additionally, GPU and CUDA were utilized to accelerate the training process. The specific configurations of the training environment are shown in Table 3. During the training process, the number of iterations was set to 300, the learning rate was set to 0.01, and the batch size was set to 4.

### 5.3. Performance Metrics and Experimental Results

The performance evaluation metrics selected in this study included the mean Average Precision (mAP) and Recall to measure the specific performance of the model. The specific content of the evaluation metrics was as follows.

The mAP is a key metric for measuring the overall performance of an object detection model, obtained by averaging the Average Precision (AP) of detections across various categories. The level of mAP directly reflects the accuracy of the model in identifying targets across different categories and is an important criterion for evaluating the effectiveness of a model. The higher the mAP, the higher the detection accuracy and the stronger the performance of the model. The corresponding calculation formulas are shown as Formulas (23) and (24):(23)$$mAP=\frac{1}{c}\sum_{i=1}^{c}AP_{i}$$
(24)$$AP=\int_{0}^{1}p\left(r\right)dr$$

In this context, m represents the number of target classification categories in the dataset, c refers to the number of categories detected within the dataset, p(r) is the Precision–Recall (P–R) curve, and AP is the area under the P–R curve for a particular category. mAP50 is an important performance metric in object detection, which is the mean AP at an Intersection over Union (IoU) threshold of 0.5. It reflects the model’s assessment of the matching degree between the predicted bounding box and the true bounding box. When the IoU reaches 0.5, meaning the predicted box and the true box overlap by more than half of their respective areas, the calculated AP at this point can provide an intuitive understanding of the model’s performance. However, *mAP*50-95 is a more comprehensive evaluation that calculates the AP at IoU thresholds ranging from 0.5 to 0.95, with steps of 0.05, and then takes the mean of these AP values. This assessment method can more delicately reflect a model’s performance across different IoU thresholds, offering richer information for the optimization and adjustment of the model.

Recall reflects a model’s ability to identify all actual existing target objects, indicating the proportion of the total number of actual target objects that the model has successfully detected. The formulas for calculating Recall are shown as Formulas (25) and (26):(25)$$\mathrm{Recall}=\frac{\mathrm{TP}}{\mathrm{TP}+\mathrm{FN}}$$
(26)$$\mathrm{Precision}=\frac{\mathrm{TP}}{\mathrm{TP}+\mathrm{FP}}$$

TP represents the number of true positives (the number of targets correctly identified by the model), while FN stands for the number of false negatives (the actual targets that the model failed to detect). The recall and precision for each category can be plotted on a Precision–Recall (P–R) curve, and the area under this curve is known as the Average Precision (AP).

We trained the model on the infrared and visible fusion image dataset, ivifdata. Table 4 presents the comparative validation results between the improved model of this study and the YOLOv8 model.

From Table 4, it can be observed that the improved model of this study, trained on the ivifdata dataset, achieved Recall, mAP50, and mAP50-95 performances of 98.7%, 98.5%, and 93%, respectively. Compared to YOLOv8n, the improvements were by 2.3%, 1.4%, and 8.2%, respectively.

Figure 23 presents a visual comparison of the 300-epoch training between the YOLOv8 network model based on the ivifdata dataset and our algorithm.

Figure 24 illustrates the comparative detection results between the original YOLOv8 network structure and the improved YOLOv8 network structure, as presented in this paper, using the ivifdata dataset. In the (b) group of comparative detection images, the YOLOv8 network exhibited false positives, misidentifying wall murals as stop signs, and missed detections, failing to recognize the tricycle and its driver, as well as not detecting some pedestrians and bicycles on the right side of the image. In the (c) group of comparative images, the YOLOv8 network missed the detection of a bicycle. In the fourth group of comparative images, there was a misclassification where a car was incorrectly identified as an airplane.

Integrating the data and comparative images, the improved YOLOv8 network model in this paper demonstrates superior performance when detecting infrared and visible fusion images. Although the targets in the fusion images, such as pedestrians, bicycles, and cars, contained both infrared and visible features, the introduction of the MSCS module significantly advanced the feature extraction capabilities of the improved YOLOv8 model. As a result, the recognition and detection rates of the improved model were higher than those of the original YOLOv8 model. This enhancement is attributed to the model’s ability to better leverage the combined characteristics of the fusion images, leading to improved accuracy and detection rates.

By utilizing the Udacity Self-Driving Car dataset and the ivifdata infrared-visible fusion image dataset for training and validation, the YOLOv8 algorithm, improved with the proposed MSCS module in this study, showed an enhancement in its accuracy and precision of target recognition. After the fusion of infrared and visible images, the target features in the images underwent significant changes. However, the improved YOLOv8 network in this study was still able to maintain high accuracy and had a lower rate of false positives. This suggests that the MSCS module effectively enhances the model’s ability to handle complex features in fused images, contributing to better detection performance.

## 6. Conclusions

To counteract the effects of varying illumination on image fusion, an adaptive illumination perception mechanism that incorporates attention to lighting conditions was proposed. This mechanism enables the fusion process to dynamically adjust, preserving more detailed features in the final fused images across different lighting scenarios. Furthermore, to optimize target detection and tracking post-fusion, an innovative MSCS module was integrated into the YOLOv8 framework. This integration led to notable improvements in the network’s performance metrics, with increases of 2.3% in Recall, 1.4% in mAP50, and 8.2% in mAP50-95. Additionally, a novel dataset, ivifdata, comprising fused infrared and visible images, was created to train and test the improved YOLOv8 model.

The comparative experiments demonstrated that the upgraded model not only achieved higher accuracy and robustness in target recognition but also exhibited a lower incidence of false positives when dealing with features that differed from those typically found in visible images. In the future, we will further refine the algorithms to enhance their real-time capabilities. Considering that the test dataset scenarios in this paper were not diverse enough, we will collect more datasets from various environments to train the algorithms.

## Figures and Tables

**Figure 1 sensors-24-06025-f001:**
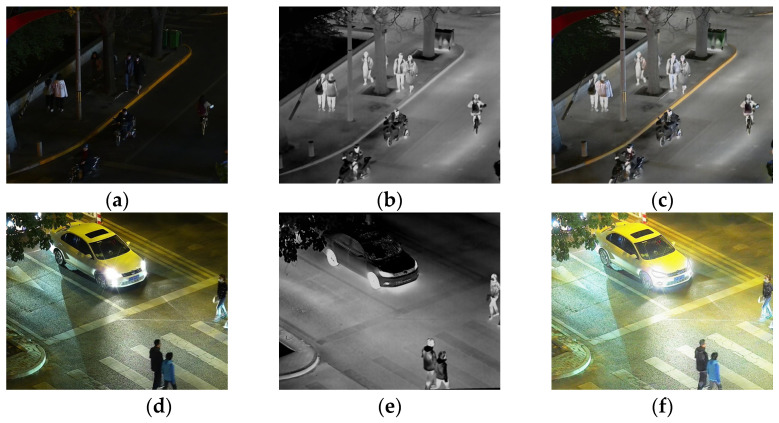
(**a**,**d**) Visible images, (**b**,**e**) infrared images, and (**c**,**f**) fused images.

**Figure 2 sensors-24-06025-f002:**
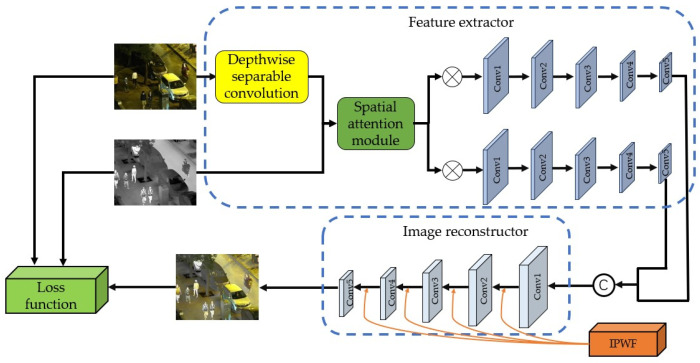
Infrared and visible image fusion network architecture.

**Figure 3 sensors-24-06025-f003:**
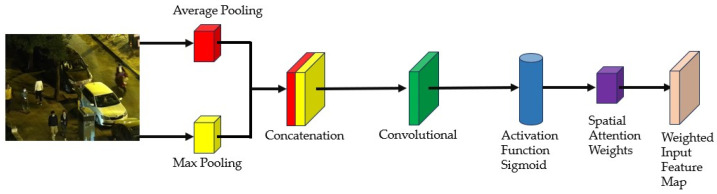
Spatial attention mechanism.

**Figure 4 sensors-24-06025-f004:**
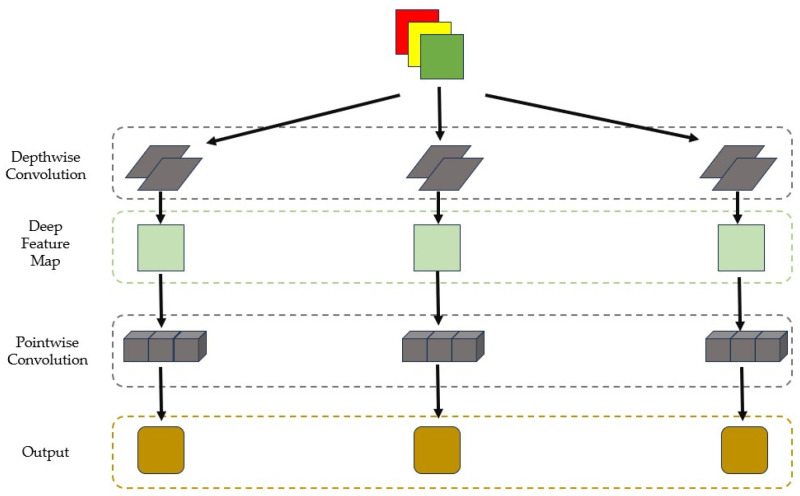
Depthwise separable convolution.

**Figure 5 sensors-24-06025-f005:**
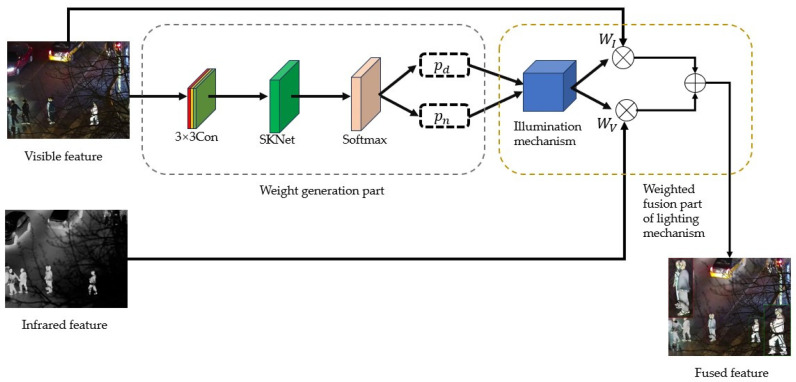
IPWF module.

**Figure 6 sensors-24-06025-f006:**
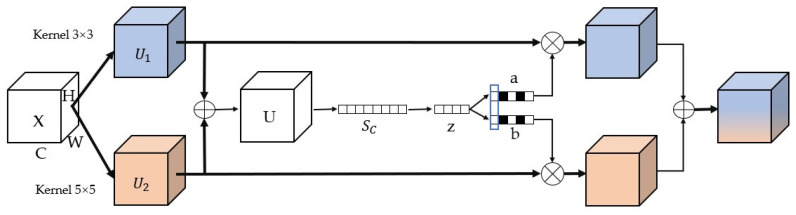
SKNet network.

**Figure 7 sensors-24-06025-f007:**
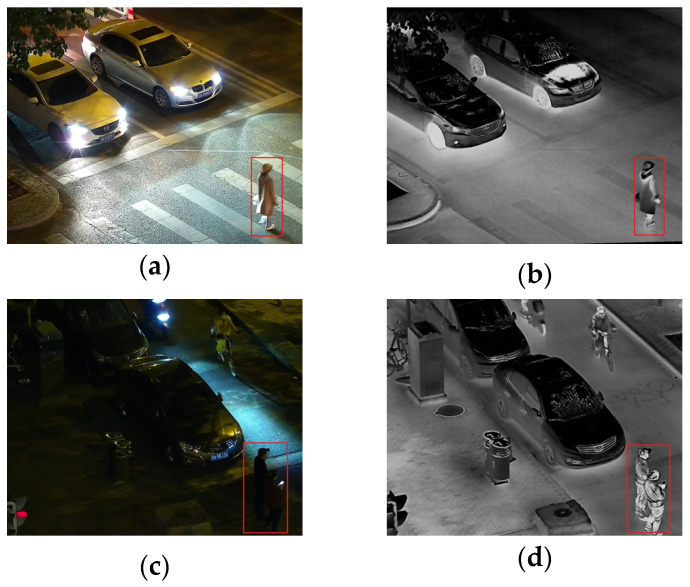
Night-time environments with strong light illumination in both visible and infrared images. (**a**,**c**) Visible images; (**b**,**d**) Infrared images.

**Figure 8 sensors-24-06025-f008:**
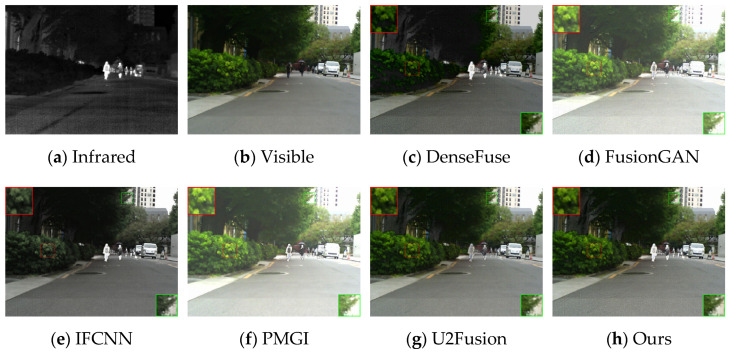
Comparative results of daytime scenarios based on the MARS (00556D) dataset.

**Figure 9 sensors-24-06025-f009:**
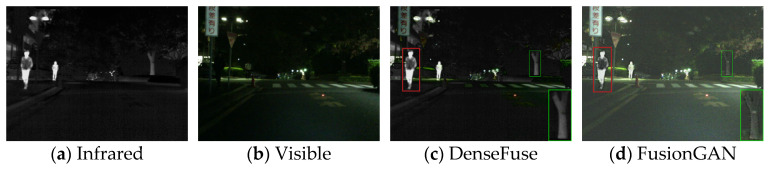
Comparative results of night-time scenarios based on the MARS (00881N) dataset.

**Figure 10 sensors-24-06025-f010:**
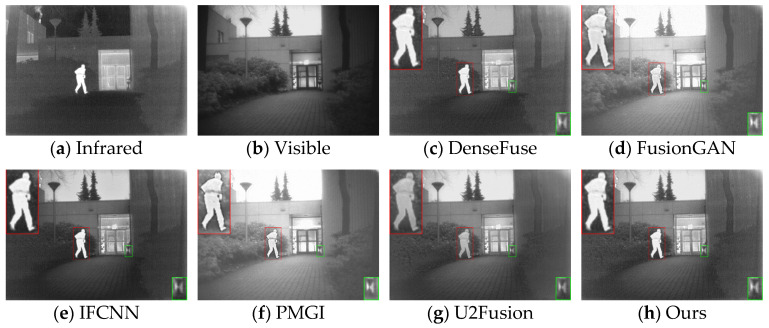
Comparative results based on the TNO (17) dataset.

**Figure 11 sensors-24-06025-f011:**
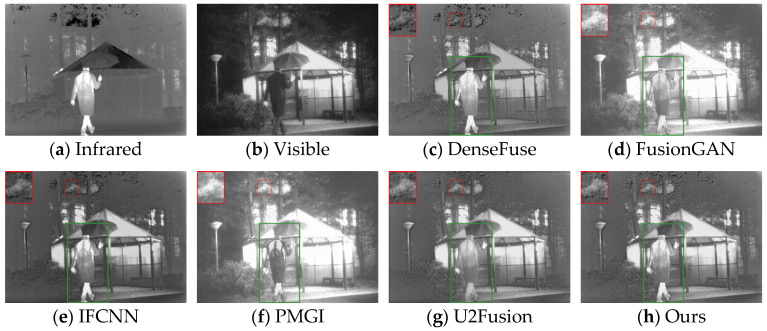
Comparative results based on the TNO (18) dataset.

**Figure 12 sensors-24-06025-f012:**
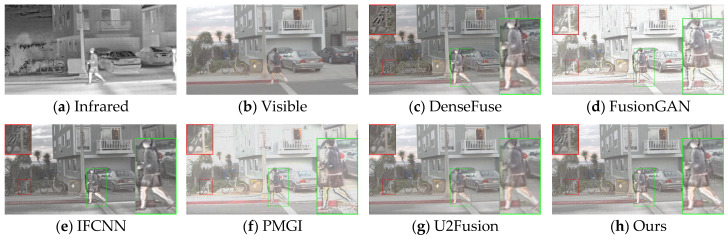
Comparative results based on the RoadScene (100) dataset.

**Figure 13 sensors-24-06025-f013:**
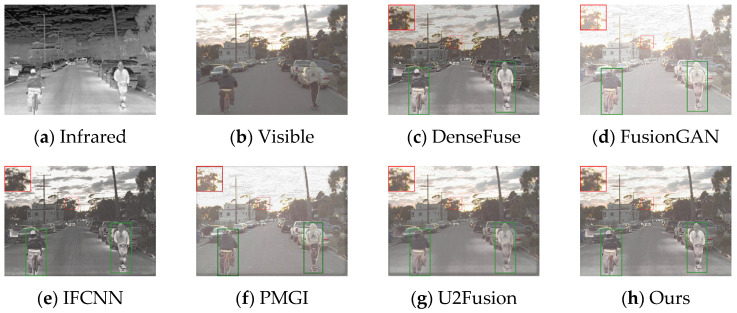
Comparative results based on the RoadScene (108) dataset.

**Figure 14 sensors-24-06025-f014:**
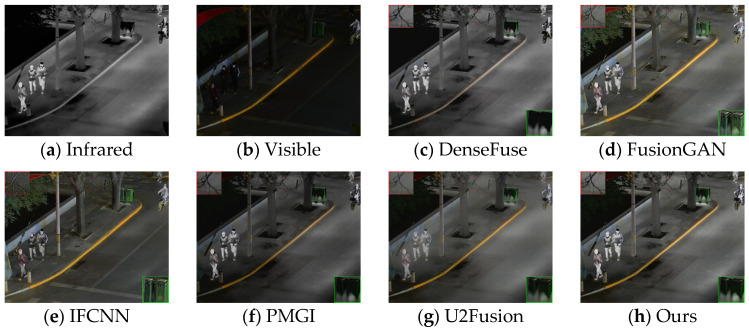
Comparative results based on the LLVIP (040223) datasets.

**Figure 15 sensors-24-06025-f015:**
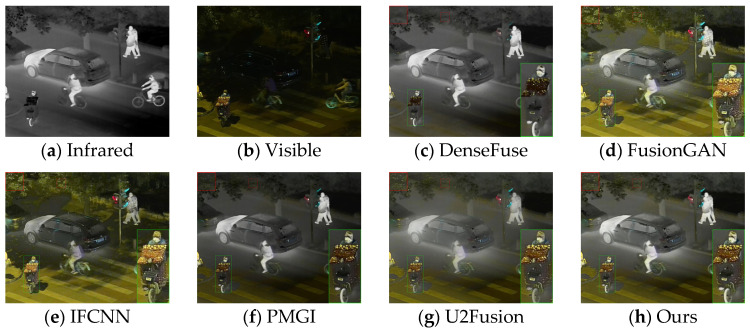
Comparative results based on the LLVIP (091079) datasets.

**Figure 16 sensors-24-06025-f016:**
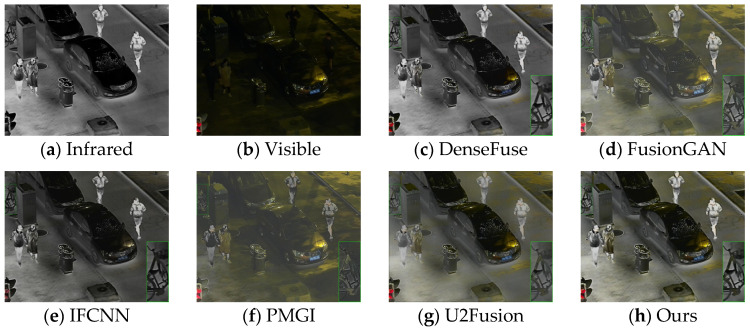
Comparative results based on the LLVIP (230114) datasets.

**Figure 17 sensors-24-06025-f017:**
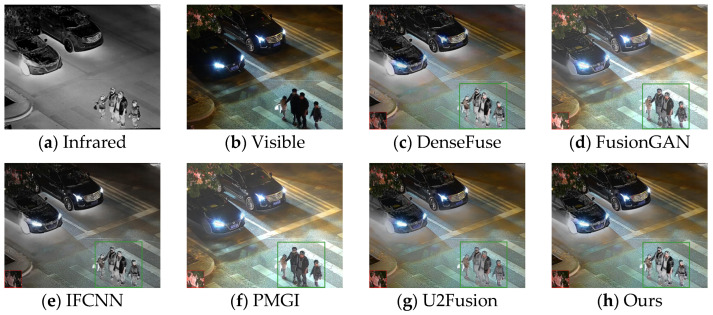
Comparative results based on the LLVIP (260199) datasets.

**Figure 18 sensors-24-06025-f018:**
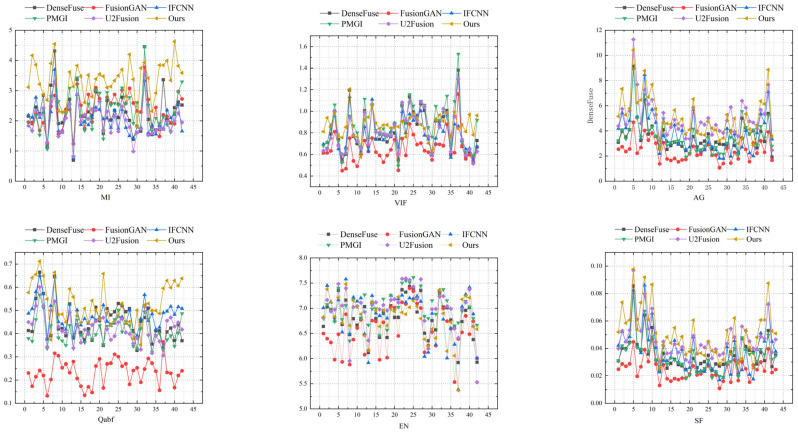
Evaluation metric line chart.

**Figure 19 sensors-24-06025-f019:**
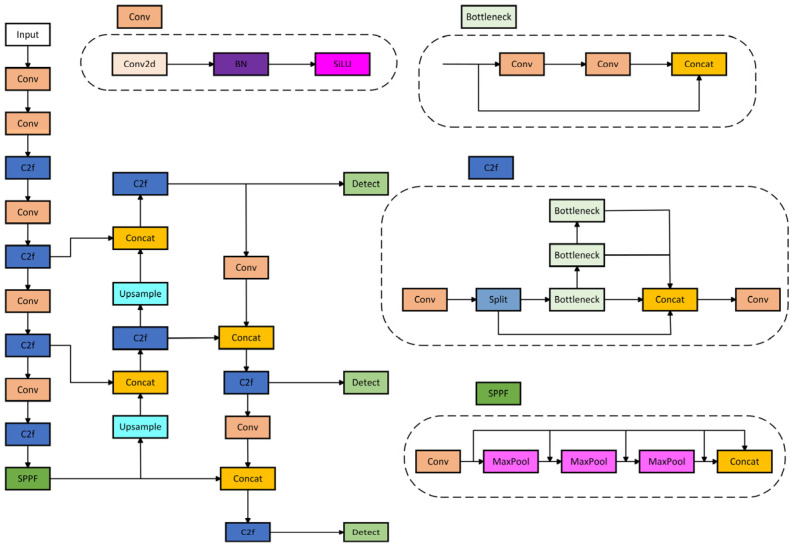
YOLOv8 network.

**Figure 20 sensors-24-06025-f020:**
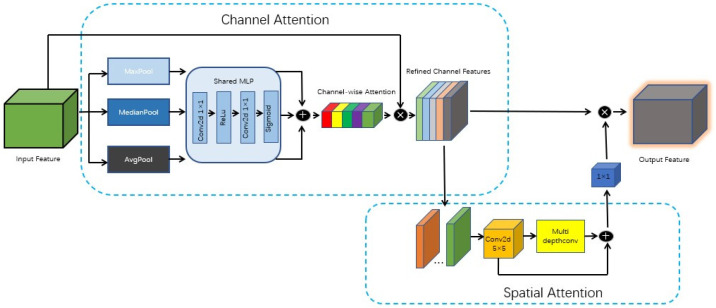
MSCS module diagram.

**Figure 21 sensors-24-06025-f021:**
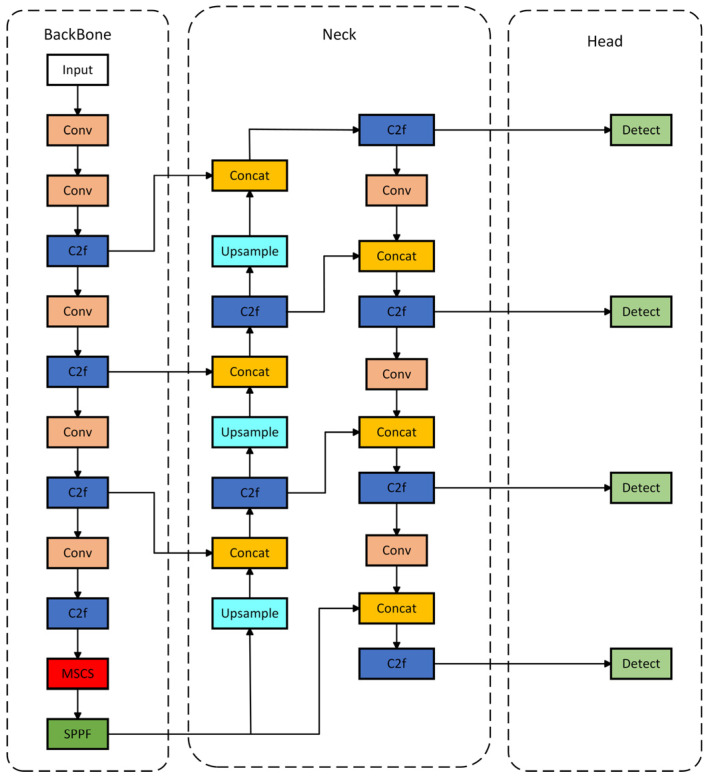
Diagram of the Improved YOLOv8 Network Structure.

**Figure 22 sensors-24-06025-f022:**
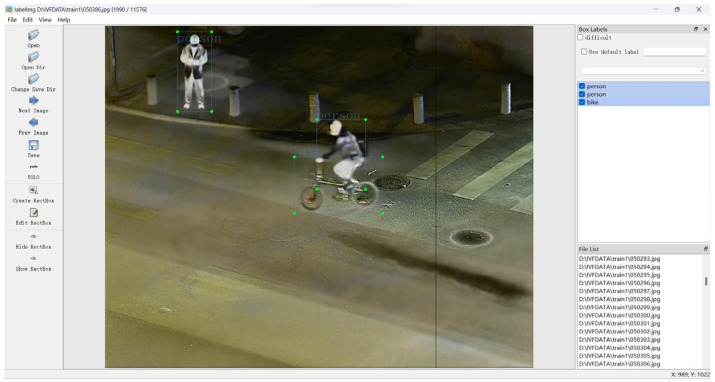
Dataset annotation.

**Figure 23 sensors-24-06025-f023:**
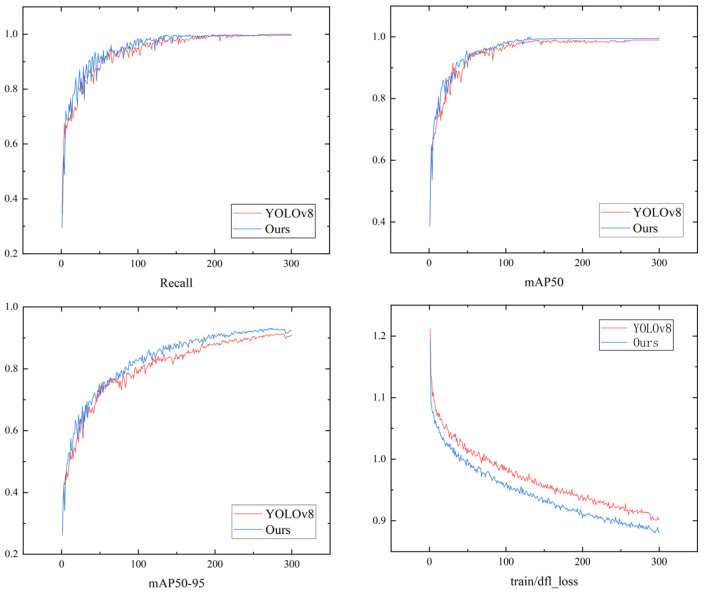
Visual comparison of the training metrics including Recall, mAP50, mAP50-95, and train/dfl_loss for the YOLOv8 network model and the algorithm presented in this paper, based on the ivifdata dataset.

**Figure 24 sensors-24-06025-f024:**
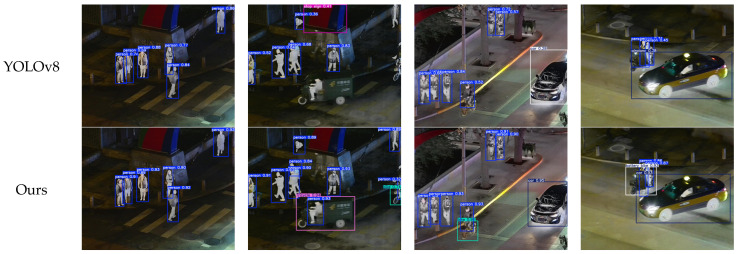
Comparative detection results on the ivifdata dataset.

**Table 1 sensors-24-06025-t001:** Experimental environment configuration.

Name	Configuration Information
Operating System	Windows11
Programming Language	Python3.8.9
Framework	Pytorch1.9.0+CUDA11.1
CPU	Intel Core i5-10210U (Intel Corporation, Santa Clara, CA, USA, equipment purchased through official channels)
GPU	NVIDIA GeForce MX350 (NVIDIA Corporation, Santa Clara, CA, USA, equipment purchased through official channels)
RAM	16 G

**Table 2 sensors-24-06025-t002:** Objective Evaluation Metric Comparison.

	MI	VIF	AG	*Qabf*	EN	SF
DenseFuse	2.3020	0.8175	3.5600	0.4457	6.8193	0.0352
FusionGAN	2.3352	0.6541	2.4210	0.2341	6.5580	0.0246
IFCNN	2.0527	0.7869	3.8280	0.4805	6.8539	0.0508
PMGI	2.3521	0.8692	3.5981	0.4117	7.0180	0.0343
U2Fusion	2.0102	0.8197	5.0233	0.4263	6.9967	0.0465
Ours	3.3576	0.8714	5.1110	0.5281	6.8143	0.0577

**Table 3 sensors-24-06025-t003:** YOLOv8 Experimental Environment Configuration.

Name	Configuration Information
Operating System	Windows11
Programming Language	Python3.9
Framework	Pytorch2.3.1+CUDA11.8
CPU	13th Gen Intel(R) Core (TM)i7-13700F (Intel Corporation, Santa Clara, CA, USA, equipment purchased through official channels)
GPU	NVIDIA GeForce RTX3060 (NVIDIA Corporation, Santa Clara, CA, USA, equipment purchased through official channels)
CPU RAM	16 G
GPU RAM	19.9 G

**Table 4 sensors-24-06025-t004:** Model comparison data based on the ivifdata dataset.

Category	YOLOv8n			Ours		
	Recall	mAP50	mAP50-95	Recall	mAP50	mAP50-95
all	0.964	0.971	0.833	0.987	0.985	0.915
person	0.961	0.973	0.807	0.997	0.993	0.94
moto	0.978	0.975	0.781	0.98	0.988	0.792
battery bike	0.951	0.987	0.808	0.975	0.995	0.941
bike	0.944	0.963	0.782	0.989	0.994	0.935
car	0.996	0.995	0.902	0.995	0.995	0.943
truck	0.98	0.995	0.901	1	0.995	0.976
fire	0.978	0.98	0.748	0.968	0.995	0.887
dog	0.97	0.955	0.971	1	0.995	0.995
tricycle	0.98	0.965	0.955	0.985	0.995	0.995
bus	0.991	0.995	0.632	0.991	0.995	0.847

## Data Availability

Due to the nature of this research, the participants of this study did not agree for their data to be shared publicly, so supporting data are not available.

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
