# Peer review of "Target Recognition Based on Infrared and Visible Image Fusion and Improved YOLOv8 Algorithm"

_sensors, 2024, doi:10.3390/s24186025_

Round 1
Reviewer 1 Report
Comments and Suggestions for Authors
The manuscript proposed an improved infrared and visible image fusion method and an improved YOLOv8 model. The work has certain novelty, however, the method description and the experiments are still not enough, as detailed below:
1. In the Introduction section, the summarized main contributions lack a more prominent description, attention mechanisms, depth-wise separable convolution are commonly adopted in many scenarios, What are the key issues that need to be addressed in introducing these modules? In addition, "imporved YOLO network", the description is too bland. What is the specific improvement?
2. For the fusion network, many descriptions need to be more detailed and accurate.
1) In Fig.3, why is there a "Global Pooling" as it is not used?
2) In Fig.4, why is it named "Factorized Convolution"? Please add the references.
3) The contents of paragraph 171-176 and paragraph 177-185 are full of contradictions and redundency, please identify the advantages and disadvantages of spatial attention.
4) Some identifiers are unclear in the description of Section 2.4. Line 249, "K" is not defined; Formula(8), 𝛾𝜔 and 𝛽𝜔 are confused.
5) For 2.5 Loss Function, is the light intensity strongly related to day and night, is there weak light during the day, strong light at night, how to deal with these situations?
6) In Formula (11), why 𝜑(𝑦) can normalize 𝜌?
3. In the experiments, many component modules lack a comprehensive analysis, such as fusion module, attention module, weight learning module.
4. The manuscript need to further polish and modify.
1) lines 74-75 and lines 78-79 are repetitive sentences.
2) line 95, "a" should be "an".
3) lines 140-142, lack references of listed attention mechanisms.
Reviewer 2 Report
Comments and Suggestions for Authors
The reviewer has provided the review report
Please see the attached file.

Reviewer 3 Report
Comments and Suggestions for Authors
This paper addresses a critical challenge in the fusion of infrared and visible images, specifically the impact of lighting variations on the fusion process. The authors propose an adaptive illumination perception fusion mechanism that is integrated into an infrared and visible image fusion network. This approach is innovative and timely, given the growing demand for robust image fusion techniques in various applications, including surveillance, military, and autonomous systems.
(1)The paper would be more impactful if it included additional technical details about the adaptive illumination perception fusion mechanism.
(2)The paper could be improved by discussing potential real-world applications of the proposed fusion mechanism. Examples of how this method could be applied in surveillance, autonomous vehicles, or other fields would provide context and illustrate the practical significance of the research.
Can be improved
Round 2
Reviewer 1 Report
Comments and Suggestions for Authors
I have no further comments on the revised manuscript.
Comments on the Quality of English LanguageThere are still some grammatical errors in the revised manuscript.